# Modeling 5-FU-Induced Chemotherapy Selection of a Drug-Resistant Cancer Stem Cell Subpopulation

**Amra Ramović Hamzagić** [1,2], **Danijela Cvetković** [1,2,*], **Marina Gazdić Janković** [1,2], **Nevena Milivojević Dimitrijević** [3], **Dalibor Nikolić** [3,4], **Marko Živanović** [3], **Nikolina Kastratović** [1,2], **Ivica Petrović** [5], **Sandra Nikolić** [1,2], **Milena Jovanović** [6], **Dragana Šeklić** [3], **Nenad Filipović** [4,7] and **Biljana Ljujić** [1,2]

1. Faculty of Medical Sciences, Department of Genetics, University of Kragujevac, 34000 Kragujevac, Serbia; ramovicamra@gmail.com (A.R.H.); marinagazdic87@gmail.com (M.G.J.); n_kastratovic@outlook.com (N.K.); sandranikolic72@yahoo.com (S.N.); bljujic74@gmail.com (B.L.)
2. Serbia for Harm Reduction of Biological and Chemical Hazards, Faculty of Medical Sciences, University of Kragujevac, 34000 Kragujevac, Serbia
3. Institute for Information Technologies Kragujevac, University of Kragujevac, Liceja Kneževine Srbije 1A, 34000 Kragujevac, Serbia; nevena.milivojevic@uni.kg.ac.rs (N.M.D.); markovac85@kg.ac.rs (D.N.); marko.zivanovic@uni.kg.ac.rs (M.Ž.); dragana.seklic@uni.kg.ac.rs (D.Š.)
4. Bioengineering Research and Development Center (BioIRC), Prvoslava Stojanovica 6, 34000 Kragujevac, Serbia; fica@kg.ac.rs
5. Faculty of Medical Sciences, Department of Pathophysiology, University of Kragujevac, 34000 Kragujevac, Serbia; i.petrovic@medf.kg.ac.rs
6. Faculty of Sciences, University of Kragujevac, Radoja Domanovića 12, 34000 Kragujevac, Serbia; milena.jovanovic@pmf.kg.ac.rs
7. Faculty of Engineering, University of Kragujevac, Sestre Janjić 6, 34000 Kragujevac, Serbia
* Correspondence: c_danijela@yahoo.com or danijela.cvetkovic@uni.kg.ac.rs

**Abstract:** (1) Background: Cancer stem cells (CSCs) are a subpopulation of cells in a tumor that can self-regenerate and produce different types of cells with the ability to initiate tumor growth and dissemination. Chemotherapy resistance, caused by numerous mechanisms by which tumor tissue manages to overcome the effects of drugs, remains the main problem in cancer treatment. The identification of markers on the cell surface specific to CSCs is important for understanding this phenomenon. (2) Methods: The expression of markers CD24, CD44, ALDH1, and ABCG2 was analyzed on the surface of CSCs in two cancer cell lines, MDA-MB-231 and HCT-116, after treatment with 5-fluorouracil (5-FU) using flow cytometry analysis. A machine learning model (ML)–genetic algorithm (GA) was used for the in silico simulation of drug resistance. (3) Results: As evaluated through the use of flow cytometry, the percentage of CD24-CD44+ MDA-MB-231 and CD44, ALDH1 and ABCG2 HCT-116 in a group treated with 5-FU was significantly increased compared to untreated cells. The CSC population was enriched after treatment with chemotherapy, suggesting that these cells have enhanced drug resistance mechanisms. (4) Conclusions: Each individual GA prediction model achieved high accuracy in estimating the expression rate of CSC markers on cancer cells treated with 5-FU. Artificial intelligence can be used as a powerful tool for predicting drug resistance.

**Keywords:** cancer stem cells; chemotherapy resistance; machine learning model

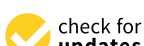



## 1. Introduction

Despite ongoing breakthroughs in oncology, malignant diseases continue to be the leading cause of morbidity and mortality worldwide [1]. Recent years have seen significant advancements in immunotherapy, gene therapy, and other treatment modalities [2]. However, chemotherapy remains a crucial component in the treatment of many malignant diseases [3]. Despite its use in combination with surgery and radiotherapy, chemotherapy

has limited efficacy and is associated with numerous unwanted side effects. Drug resistance, caused by various mechanisms employed by tumor tissue to overcome the effects of drugs, remains a major challenge, and it is believed that disease progression is often due to the emergence of treatment resistance [2]. One mechanism contributing to treatment survival and disease relapse is the presence of cancer stem cells (CSCs) within the tumor tissue. These cells possess pluripotent properties [4]. CSCs are defined as a subpopulation of tumor cells with the capacity for self-renewal and differentiation to drive the initiation, progression, metastasis, and recurrence of tumors [5]. There are CSCs that are part of the tissue milieu of the tumor even before the application of therapy, but today, it is well known that even well-differentiated cells are subject to further differentiation in the sense of a shift towards the pluripotent part of the cell spectrum, when CSCs subsequently arise, sometimes paradoxically even in response to the oncological treatment applied (chemo or radiotherapy) [4]. Today, markers have even been discovered on the cell surface that can identify CSCs, which can be an important therapeutic target, such as CD44 (CD44high) combined with the (near-)absence of CD24 (CD24low) [6]. The presence of undifferentiated CD44+CD24−/low tumor cells is an unfavorable prognostic marker in patients with breast cancer, and the increased proportion of CD44+CD24−/low cells is an indicator of probable metastases in the lymph nodes [7]. The peritumor environment and its modification during the application of therapy also play a very important and active role in the processes of resistance to treatment [8]. After the eradication of most tumor cells, only CSCs remain in the tissue milieu of the tumor. Being resistant to therapy, CSCs can not only survive oncological treatment but also continue to proliferate unhindered, creating relapses and distant metastatic deposits resistant to the initial therapy [9]. Mathematical modeling, in silico experiments, and computer simulations are being integrated into biomedicine and clinical practice. Computer systems are essential in medicine, machine learning, and personalized therapy [10]. The integration of computer modeling and simulations has become crucial in scientific research, offering a cost-effective and efficient alternative to complex and time-consuming experiments. Machine learning (ML) models, such as the genetic algorithm (GA), are employed to simulate real processes [11]. GA utilizes biologically inspired mathematical operators like mutation, crossover, and selection based on Charles Darwin's theory of natural evolution [12]. This study aimed to identify ways to enhance chemotherapy treatments and overcome drug resistance by utilizing a mathematical model of chemotherapy. The researchers discovered valuable insights that could contribute to improving the application of anticancer therapies, particularly in the context of precision medicine.

## 2. Materials and Methods

### 2.1. Cell Culturing and Chemotherapy Treatment

The in vitro experiments were conducted on two distinct human cancer cell lines: HCT-116, which is derived from human colorectal carcinoma, and MDA-MB-231, originating from human breast adenocarcinoma. These cell lines were chosen for their relevance to their respective cancer types and their well-documented biological behaviors. HCT-116 cells are characterized by specific genetic mutations and molecular pathways that are emblematic of colorectal cancer, while MDA-MB-231 cells are representative of triple-negative breast cancer, which is known for its aggressive nature and lack of hormonal receptors. Each cell line presents unique challenges in terms of treatment and reflects the heterogeneity of cancer biology, which is critical for the evaluation of chemotherapeutic efficacy and the analysis of cancer stem cell (CSC) marker expression. The cells were cultured under standard conditions (ECACC—European Collection of Authenticated Cell Cultures), and following 24 h of seeding, they were exposed to the chemotherapeutic agent 5-fluorouracil (5-FU) to assess the impact on CSC marker expression. For chemotherapy treatments, the cells were exposed to 5-FU at a clinically relevant concentration, which was determined based on prior dose–response studies (62.5 μM). The treatment schedule involved administering 5-FU for a period reflective of standard clinical protocols, with exposure times ranging

from 24 to 52 h to assess short-term and prolonged effects. Untreated control cells were maintained under identical conditions without 5-FU exposure at each time point to serve as baseline controls for comparison.

### 2.2. Flow Cytometry Analysis

In order to screen the expression of various cancer stem cell surface markers, the cells were cultured in 6-well dishes. After 24 h of seeding, the cells were treated with the chemotherapy agent 5-FU. For the flow cytometric analysis, $1 \times 105$ cells were harvested at 24, 33, 43-, and 52 h post treatment and incubated with anti-human CD24, CD44, ALDH1, and ABCG2 monoclonal antibodies conjugated with allophycocyanin (APC), peridinin chlorophyll protein (PerCP), phycoerythrin (PE), or fluorescein isothiocyanate (FITC) (all from BD Biosciences, San Jose, CA, USA), following the manufacturer's instructions. It is important to note that for the detection of intracellular ALDH1A1, which is not a surface protein, the cells were permeabilized using a fixation/permeabilization solution kit (BD Biosciences) following the provided protocol before incubation with the antibodies. This allows for the intracellular staining of ALDH1A1, enabling its detection alongside surface markers. Flow cytometric analysis was conducted on a BD Biosciences FACSCalibur, and the data were analyzed with commercial software, Flowing. For the control cells, identical procedures were followed without the addition of 5-FU to ensure the comparability of the data.

We used forward and side scatter density plots to identify the CSC population of interest and exclude debris. Forward versus side scatter (FSC vs. SSC) gating is commonly used to identify cells of interest based on size and granularity (complexity). It is suggested that forward scatter indicates cell size, whereas side scatter relates to the complexity or granularity of the cell. In our samples, with HCT-116 and MDA.MB-231 cells, this first level gating method was crucial for identifying the cells of interest. This gating strategy was also used to exclude debris as they tend to have lower forward scatter levels. Dead cells were found at the bottom left corner of the FSC vs. SSC density plot. Dead cells and cell debris are characterized by lower FSC/SSC values (lower FSC values indicate non-proliferative cells, and higher SSC values indicate granular formation during apoptosis); therefore, they are easily excluded from the live region.

### 2.3. Machine Learning Model (ML)–Genetic Algorithm (GA)

In our methodology, the machine learning component utilizes a genetic algorithm (GA) within a Genetic Programming (GP) framework, specifically using a symbolic regressor. This choice was motivated by the ability of the GP symbolic regressor to generate a mathematical function that not only represents the input data accurately but also produces an output that is comprehensible and translatable to other applications or environments. The novelty of our approach lies in the specific application of this regressor to the prediction of CSC marker expression following chemotherapy treatment. Unlike the methods described by O'Neill et al., our model incorporates a unique set of features derived from our experimental data, which includes not only the standard gene expression profiles but also a range of cellular responses to 5-FU treatment. This allows our model to simulate the biological behavior of cancer cells post-chemotherapy and predict the dynamics of CSC marker expression over time more precisely. To enhance transparency and reproducibility, we provide a detailed description of the GA's configuration in the Supplementary Materials, including population size, mutation rates, and selection mechanisms. Furthermore, we have conducted a series of validation experiments to compare the predictions of our model with empirical data, confirming the robustness and predictive power of our approach. These results are presented in the Results section and demonstrate that our machine learning model can serve as a valuable tool for simulating the complex biological processes associated with CSC behavior and drug resistance. In the GP, the mathematical function is represented as a tree, with the sheets serving as the variables or constants and the functions serving as the nodes (branching points). Nodes have the possibility to be different functions

from the list in the function set [add, sub, mult, div, sqrt, log, abs, neg, inv, max, min, sin, cos, and tan], and leaves can be determined in the terminal set for constant values of a defined range or variables. Nodes and leaves are primarily acquired randomly; crossover and mutation reproduction change them. In order to evaluate the effectiveness of the results and select the best ones for inclusion in the next iteration of the genetic algorithm, a population of children was examined after the genetic operation had been carried out. The loop was terminated when the algorithm reached the stopping threshold or the maximum number of generations. The operating principles are detailed and described by O'Neill et al. [13]. GP is not sequential or time-dependent and does not have memory. It is a simple algorithm that sets the past input values of time series in multiple points and other variables for the prediction of future value. Input data for training GA and fitting were used from experimental measurements on flow cytometry of CSC markers expression in cancer cells treated with 5-FU and without treatment. Several markers (CD24, CD44, ALDH1, and ABCG2) were measured in time (24 h, 33 h, 43 h, and 52 h). The results from measurements at 52 h were used for GA model validation. Figure 1 presents the measured data for MDA-MB-231 and HCT-116 cell lines for the input data.

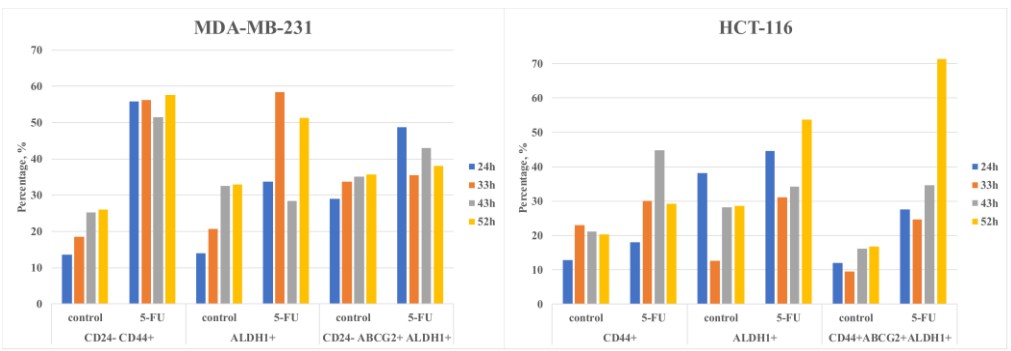

**Figure 1.** Flow cytometry of MDA-MB-231 and HCT-116 cells for input data.

*2.4. Statistical Analysis*

In this study, we utilized Statistical Package for the Social Sciences v23.0 software IBM Coro., Armonk, NY, USA (SPSS Inc.) to assess data normality using the Shapiro–Wilk test and conduct statistical analyses. Subsequently, we performed independent samples *t*-tests or Mann–Whitney U tests to compare the means between groups, depending on the assumptions of normality and homogeneity of variances. The results are expressed as mean values $\pm$ standard error (SE). A *p*-value of less than 0.05 was considered to indicate statistical significance.

## 3. Results

*3.1. CSCs Markers Analyzed by Flow Cytometry*

We evaluated the effect of 5-FU treatment on the percentage of tumor cells expressing CSC markers. Analysis was performed on viable MDA-MB-231 cells, and the percentages of dead cells during incubation with 5-FU and in controls are presented in Figure 2. Representative dot plots of CD24-CD44+, ALDH1, and ABCG2 marker expressions on the MDA-MB-231 cell line (control group and 5-FU-treated group) are available in the Supplementary Materials Figure S1. As evaluated via flow cytometry, the percentage of CD24-CD44+ MDA-MB-231 cells treated with 5-FU for 24-, 33-, 43-, and 52 h was significantly increased compared to the untreated cells (*p* < 0.001) (Figure 3A and Supplementary Figure S1A). Additionally, 5-FU-treated tumor cells showed a high rate of expression for CSC marker ALDH1 compared to their matched control cells (*p* < 0.001) (Figure 3B and Supplementary Figure S1B). Although the difference did not reach statistical significance at all time points, our analysis revealed the increased presence of the CD24-ABCG2+ ALDH1+ population after 5-FU treatment, indicating that this population was resistant to the chemotherapeutic agent (Figure 3C and Supplementary Figure S1C).

**Figure 2.** Percentage of cell death induced by 5-FU incubation. The data are presented as means $\pm$ SEM of three independent experiments. * $p < 0.01$, *** $p < 0.001$. Graphs represent cumulative data from 24 to 52 h.

In Figure 3 and Supplementary Figure S1, the time points displayed in the graphs and histograms differ due to the observed saturation of marker expression beyond this point. For instance, longer time points were included in the graphs (up to 52 h) to illustrate the overall trends, while histograms were limited to 43 h to focus on the most significant changes in marker expression. Histograms depict marker expression up to 43 h, beyond which saturation was observed without additional insights into treatment effects. Representative flow cytometry dot plots illustrating the gating strategy and the percentage of cells expressing CSC markers in MDA-MB-231 treated with 5-FU compared to untreated controls at various time points are depicted in Supplementary Figures (Figures S15 and S16).

Moreover, we detected a trend relating to a significant increase in CSC-related markers expression (CD44, ALDH1 and ABCG2) in 5-FU-treated HCT-116 cells after 24 h to 52 h (Figure 4). Only viable HCT-116 cells were analyzed, and the percentage of dead cells, both treated and untreated with 5-FU, are presented in Figure 5. Representative dot plots of CD44, ALDH1, and ABCG2 marker expression on the HCT-116 cell line (control group and the 5-FU-treated group) are featured in the Supplementary Materials (Figure S2). As shown in Figures 4A and S2A, the percentage of CD44+ HCT-116 cells significantly increased in the presence of 5-FU over 24, 33, and 43 h. Similar expression patterns were seen 52 h after treatment, but there were no significant differences between the groups. The same is observed with the ALDH1 marker as an indicator of metastasis. In particular, 5-FU treatment enhanced the expression of ALDH1 compared to untreated HCT-116 cells in a time-dependent manner (Figure 4B and Supplementary Figure S2B). We also found that the presence of CD44+ALDH1+ABCG2+ tumor cells steadily increased over time ($p < 0.001$) (Figure 4C and Supplementary Figure S2C). After 52 h of treatment, the percentage of the CSC population was higher than in the control cells by about 3.5-fold. The time points depicted in Figure 4 and Supplementary Figure S2 exhibited variations in the graphs and histograms, primarily due to the observed saturation of marker expression beyond a certain threshold. To elucidate, the graphs encompass extended time intervals to portray overarching trends, whereas the histograms are confined to 43 h, emphasizing the most impactful alterations in marker expression. Specifically, the histograms illustrate marker expression up to 43 h, as beyond this point, the saturation of marker expression did not yield further insights into the effects of the treatment. Flow cytometry dot plots showing the gating strategy used in the identification of CD44+, ALDH1+ and CD44+ABCG2+ALDH1+

HCT-116 cells treated with 5-FU 24, 33, 43, and 52 h post-treatment are presented in Supplementary Figures (Figures S17 and S18).

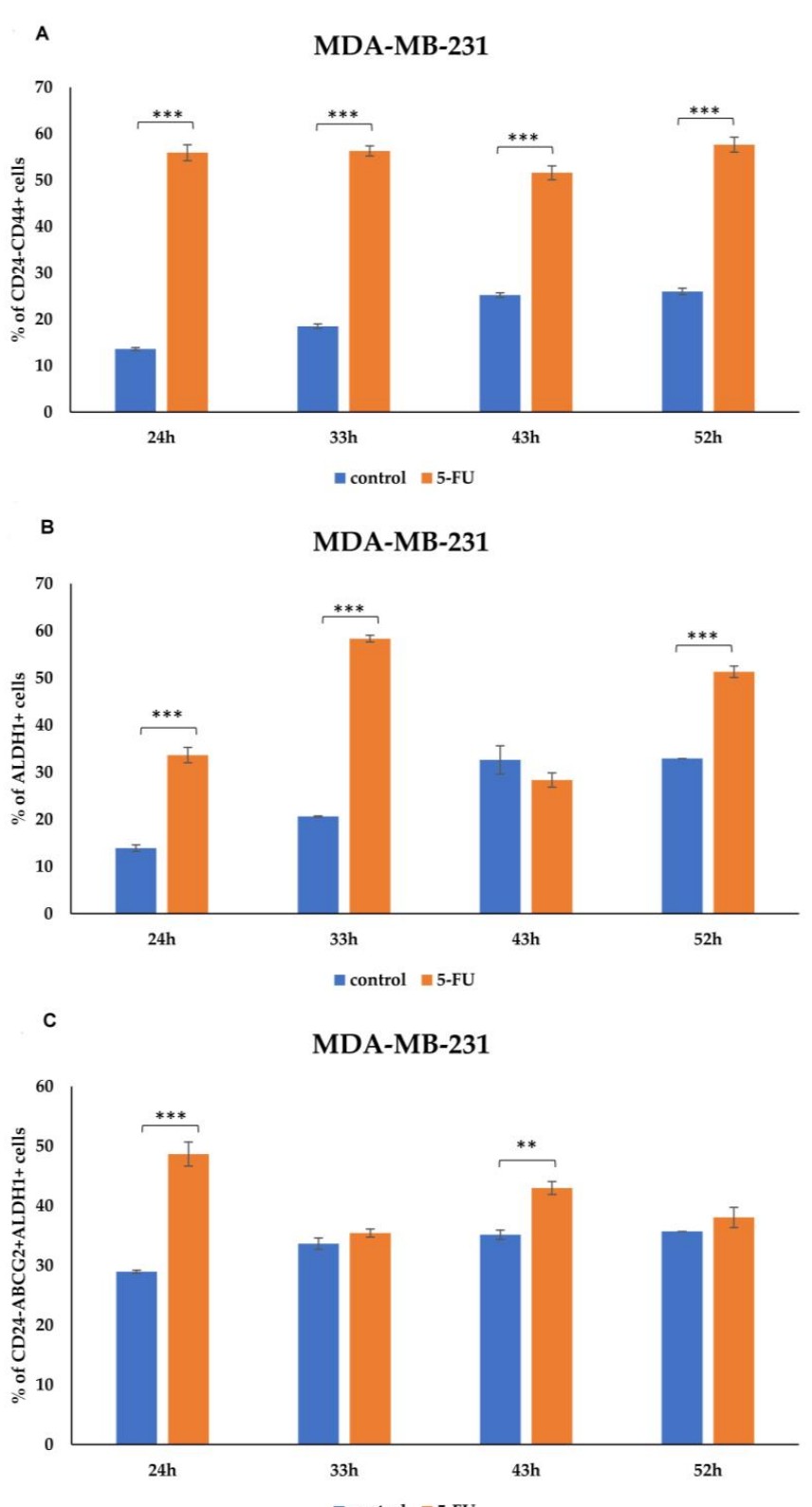

**Figure 3.** Expression rate of CSCs markers in MDA-MB-231 cells. Percentage of (**A**) CD24-CD44+, (**B**) ALDH1+, and (**C**) CD24-ABCG2+ ALDH1+ tumor cells untreated and treated with 5-FU, analyzed using flow cytometry. The data are presented as means ± SEM of three independent experiments. ** $p < 0.01$, *** $p < 0.001$. Graphs represent cumulative data from 24 to 52 h.

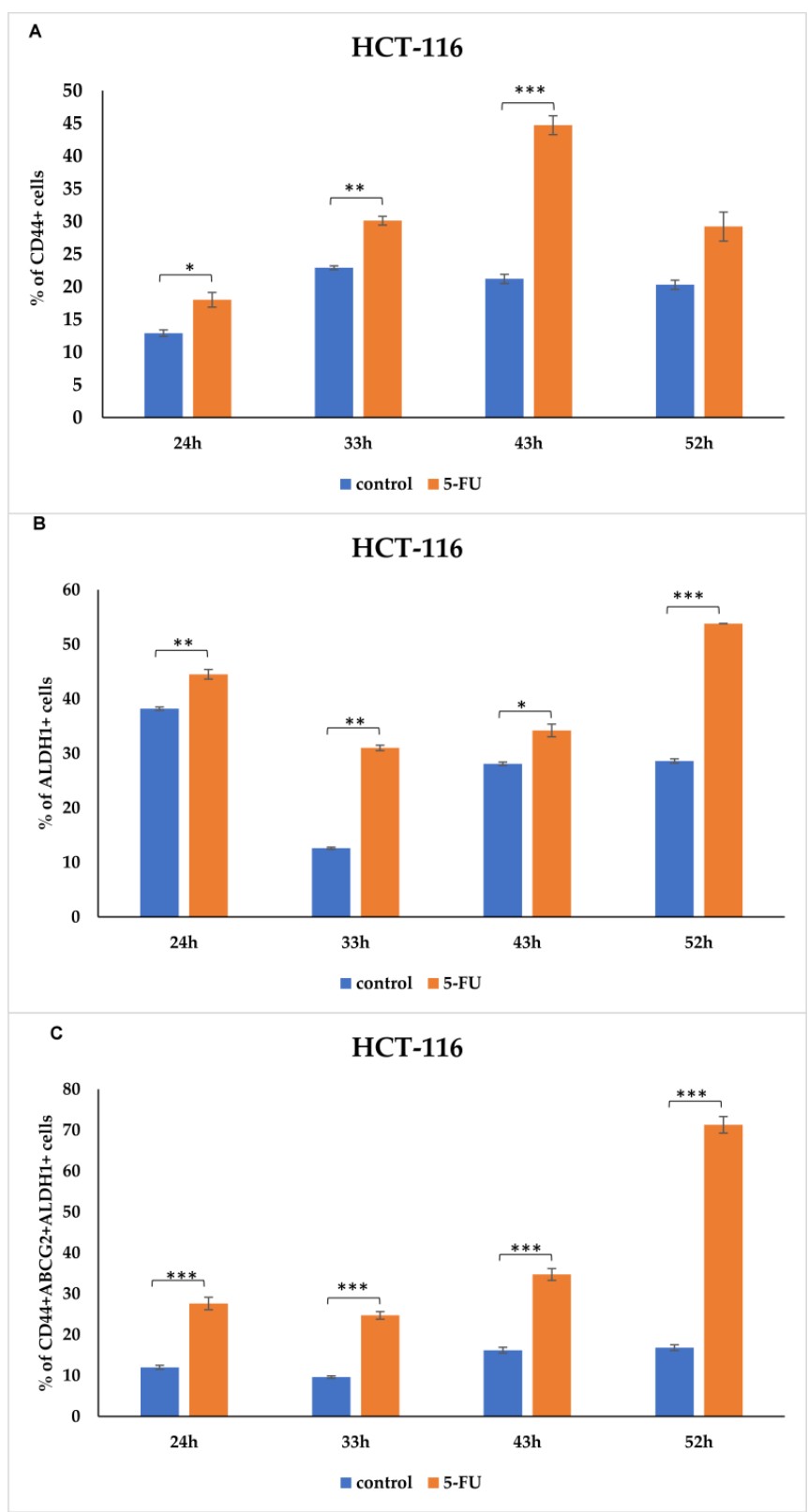

**Figure 4.** Expression rate of CSCs markers in HCT-116 cells. Percentage of (**A**) CD44+, (**B**) ALDH1+, and (**C**) CD44+ ABCG2+ ALDH1+ tumor cells untreated and treated with 5-FU, analyzed using flow cytometry. The data are presented as means $\pm$ SEM of three independent experiments. * $p < 0.05$, ** $p < 0.01$, *** $p < 0.001$. Graphs represent cumulative data from 24 to 52 h.

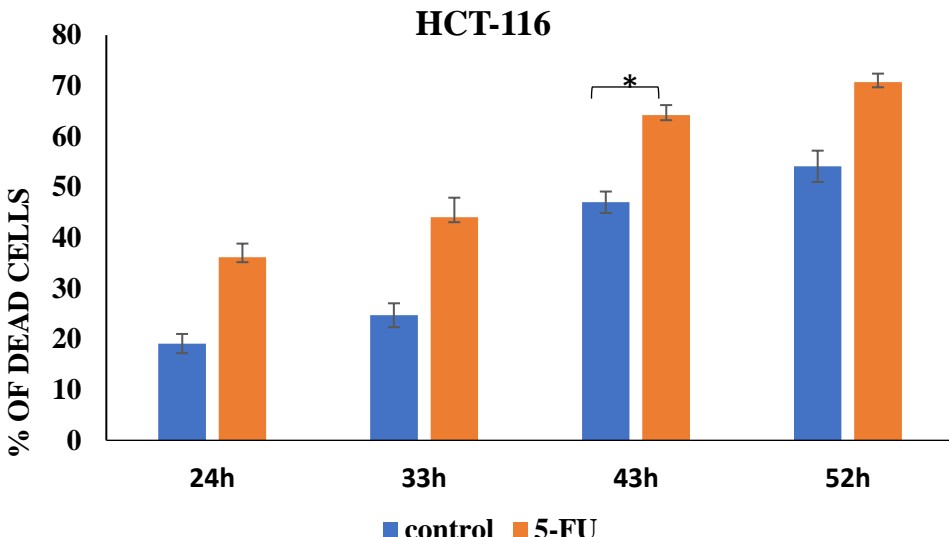

**Figure 5.** Percentage of cell death induced by 5-FU incubation. The data are presented as means $\pm$ SEM of three independent experiments. * $p < 0.01$. Graphs represent cumulative data from 24 to 52 h.

### 3.2. Genetic Algorithm (GA)

Figure 6 shows the real measured data for the MDA-MB-231 cell line 24 h–43 h (diamond dots) and the estimation of the GA (dashed curve) for a period of 24 h to 67 h. The measured data in 52 h (triangle dot) were used for the validation of the GA decision model, and follow-up predicting results of 67 h are presented as X dots on the graphics. Table 1 presents the algorithm estimation score with $R^2$.

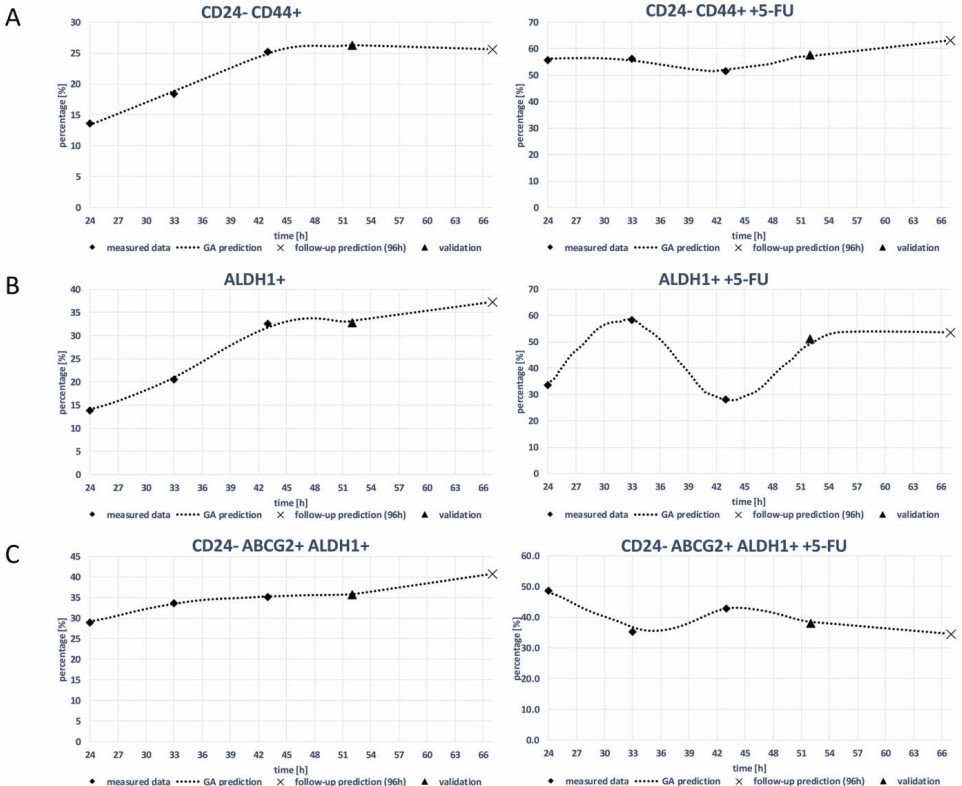

**Figure 6.** GA prediction of the expression rate of CSC markers. (**A**) CD24- CD44+, (**B**) ALDH1+, and (**C**) CD24- ABCG2+ALDH1+ on MDA-MB-231 cell line without and with 5-FU treatment (GA decision tree are presented in Figures S3–S8).

**Table 1.** $R^2$ score of the prediction.

| Model | $R^2$—Score of the Prediction |
|---|---|
| MDA-MB-231 CD24- CD44+ | 0.97 |
| MDA-MB-231 CD24- CD44+ + 5-FU | 0.98 |
| MDA-MB-231 ALDH1+ | 0.96 |
| MDA-MB-231 ALDH1+ + 5-FU | 0.98 |
| MDA-MB-231 CD24- ABCG2+ ALDH1+ | 0.95 |
| MDA-MB-231 CD24- ABCG2+ ALDH1+ + 5-FU | 0.96 |
| HCT-116 CD44+ | 0.97 |
| HCT-116 CD44+ + 5-FU | 0.97 |
| HCT-116 ALDH1+ | 0.98 |
| HCT-116 ALDH1+ + 5-FU | 0.93 |
| HCT-116 CD44+ ABCG2+ ALDH1+ | 0.96 |
| HCT-116 CD44+ ABCG2+ ALDH1+ + 5-FU | 0.95 |

Figure 7 shows the predicted expression rates of three cancer stem cell (CSC) markers, CD44+, ALDH1+, and CD44 + ABCG2+ ALDH1+, in the HCT-116 cell line with and without 5-FU treatment, as predicted through the use of a genetic algorithm (GA) decision tree. The GA decision tree is a machine learning algorithm that has been trained on a dataset of gene expression data from HCT-116 cells. The GA decision tree can be used to predict the expression of CSC markers in new HCT-116 cells based on their gene expression profile. This figure illustrates the predictive modeling results of CSC marker expression rates in HCT-116 cell lines post 5-FU treatment, as determined through the use of a genetic algorithm (GA) decision tree. The GA model, trained on gene expression data from HCT-116 cells, provides a forecast of CSC marker levels, offering insights into the potential resistance patterns post chemotherapy. Furthermore, the goodness-of-fit for the GA model predictions is quantified by the $R^2$ values, which are provided alongside each plot. For instance, an $R^2$ value of 0.96 indicates a high level of accuracy in the model's prediction of CSC marker expression at the 52 h mark. The corresponding $R^2$ values for each time point and marker combination are detailed within the figure, providing a metric for the reliability of the predictions made by the GA decision tree.

The visualizations and data presented in Figures 4 and 6 not only allow for the assessment of the predictive power of the GA model but also facilitate a deeper understanding of the dynamic changes in CSC marker expression in response to chemotherapeutic treatment.

Table 1 presents the $R^2$ scores for various models and treatment conditions on the MDA-MB-231 and HCT-116 cell lines, indicating the goodness of fit between the models and the actual data. Higher $R^2$ values suggest better model performance. In the MDA-MB-231 model, the CD24-CD44+ model exhibits a high $R^2$ score of 0.97; introducing 5-fluorouracil (5-FU) to the CD24-CD44+ model improves the $R^2$ to 0.98; the ALDH1+ model has an $R^2$ value of 0.96, while the addition of 5-FU enhances it to 0.98; and the CD24-ABCG2+ ALDH1+ model shows an $R^2$ value of 0.95, with 5-FU improving the result to 0.96. In the HCT-116 model: the CD44+ model has an $R^2$ value of 0.97; the $R^2$ value for the CD44+ model with added 5-FU is not provided; the ALDH1+ model demonstrates an $R^2$ value of 0.97, and 5-FU increases it to 0.98; the ALDH1+ model experiences a decrease in $R^2$ value (0.93) after 5-FU treatment; and the CD44+ ABCG2+ ALDH1+ model has an $R^2$ value of 0.96, with 5-FU improving the result to 0.95. Overall, the addition of 5-FU often enhances $R^2$ values, suggesting a positive impact on model accuracy in assessing marker expression on cell surfaces. However, there are exceptions, such as the decrease in $R^2$ value in the HCT-116 ALDH1+ model after 5-FU treatment.

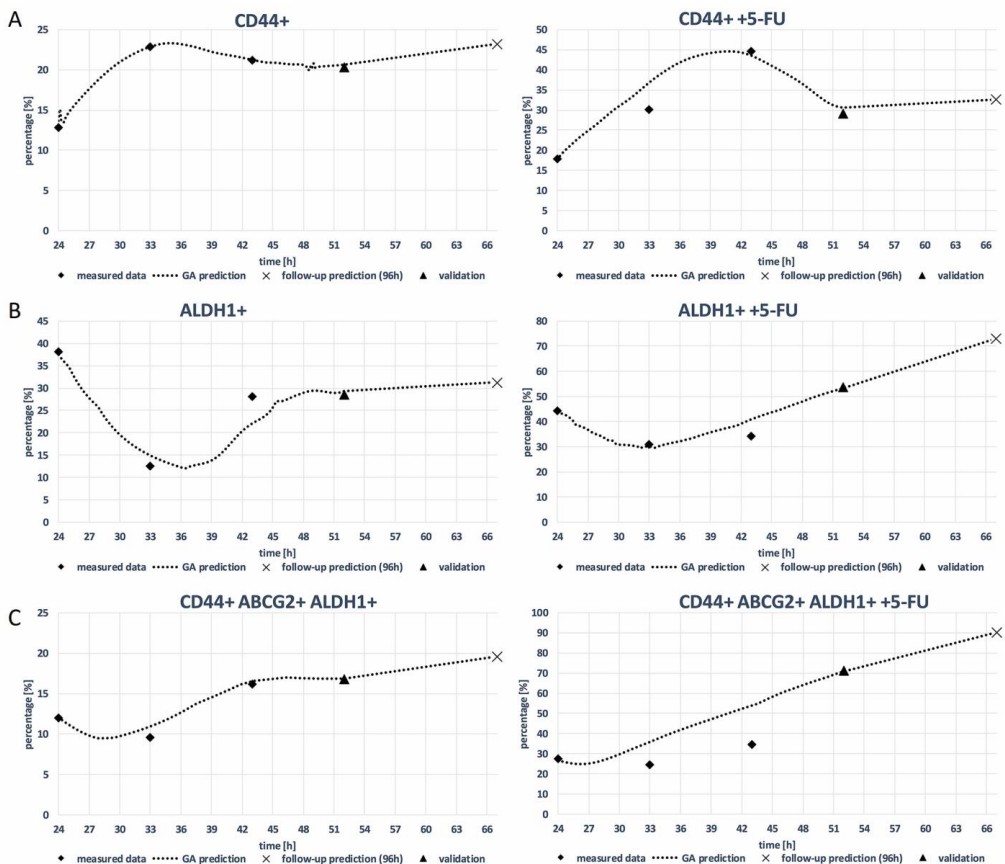

**Figure 7.** GA prediction of expression rate of CSCs markers (**A**) CD44+, (**B**) ALDH1+, and (**C**) CD44+ ABCG2+ALDH1+ on HCT-116 cell line with and without 5-FU treatment (GA decision tree was present in Figure S9–S14).

## 4. Discussion

Cancer stem cells (CSCs) are small subpopulations of cells in a tumor that have the ability to self-regenerate and produce different types of cells in the tumor. These cells are thought to be responsible for tumor recurrence and resistance to chemotherapy. One approach to examining cancer stem cells is to use immunophenotypic analysis. This involves identifying specific cell surface markers that are characteristic of cancer stem cells and can be used to distinguish them from other cells in a tumor. After separation, these cells can be further characterized at the molecular level. Cancer stem cells have the ability to self-renew, differentiate, and initiate tumor growth and metastasis. The identification of markers that are specific to CSCs is important for understanding their biology and developing targeted therapies [4,14]. Cancer stem cells (CSCs) are a crucial subpopulation within tumors, known for their ability to self-renew, differentiate, and initiate tumor growth and metastasis. The identification of cell surface markers that are characteristic of CSCs is pivotal for distinguishing them from other tumor cells and understanding their role in cancer progression and resistance to therapy. While this study focused on the expression of the markers CD24, CD44, ALDH1, and ABCG2, it is recognized that CSC biology is not limited to these markers alone. CSCs are defined by a complex profile that may include additional markers such as EpCAM, PROM1, THY1, and others. These markers collectively contribute to the multifaceted nature of CSCs and their ability to sustain tumor growth and resist chemotherapeutic agents. In this context, our analysis considered the expression of CD24-CD44+, which was significantly elevated in MDA-MB-231 cells treated with 5-FU, as a starting point. However, future studies should expand the scope to include a broader spectrum of CSC markers, thereby providing a more comprehensive understanding of the CSC phenotype and its implications for therapy. The integration of these additional markers

may also yield insights into the heterogeneity of CSCs and the dynamic nature of their expression in response to chemotherapeutic treatment. The results of this study showed that the expression of CD24-CD44+ was statistically significantly elevated in all observation points on the surface of MDA-MB-231 cells treated with 5-FU. This indicates that the CSC population was enriched after treatment with chemotherapy. Additionally, ALDH1 expression was elevated after 24, 33, and 52 h, suggesting that these cells have enhanced drug resistance mechanisms. CD24-ABCG2+ALDH1+ expression was also elevated after 24 and 43 h, indicating that these cells may have increased resistance to chemotherapy through the efflux of drugs mediated by ABCG2 [15]. In HCT-116 cells, the expression of CD44+ was statistically significantly elevated on the surface of cells treated with 5-FU after 24, 33, and 43 h. This suggests that the CSC population was also enriched after treatment with chemotherapy. ALDH1 expression was elevated at all observation points, indicating that these cells have enhanced drug resistance mechanisms. CD44+ALDH1+ABCG2+ expression was also elevated in all observation points, indicating that these cells may have increased resistance to chemotherapy through the efflux of drugs mediated by ABCG2 [16]. These findings suggest that the expression of CD24, CD44, ALDH1, and ABCG2 on the surface of CSCs may serve as prognostic factors for resistance to chemotherapy in cancer patients. Furthermore, the results of this study highlight the importance of developing targeted therapies that can specifically target CSCs to improve the efficacy of chemotherapy and prevent tumor recurrence [17]. Examining cancer stem cell receptors for resistance after cytostatic therapy is important for several reasons. First, cancer stem cells are the main source of tumor growth and tumor survival and are considered a key target in cancer therapy. They are characterized by a high degree of heterogeneity and plasticity, which means that they have the ability to change their phenotype and adapt to the environment. Therefore, examining the receptors on cancer stem cells for resistance after cytostatic therapy may help in understanding the mechanisms that allow the survival of cancer stem cells after therapy and further tumor growth [18]. Cytostatics are a common therapy for cancer, but not all patients are equally sensitive to these drugs. Examining the receptors on cancer stem cells can provide insight into possible causes of therapeutic resistance and help identify new therapeutic targets. Identifying markers of resistance to cancer stem cells can help in the development of personalized cancer therapy, which would be based on the characteristics of an individual patient's tumor. Predicting resistance to cytostatics is very important in cancer medicine and research. If the tumor is resistant to cytostatics, treatment may be less effective and have many severe side effects; therefore, it is important to recognize patients who might not respond to cytostatic therapy [19]. Predicting resistance to cytotoxic drugs can help doctors choose the best treatment for a patient. In summary, the examination of receptors on cancer stem cells for resistance after cytostatic therapy is important for understanding the mechanisms that enable the survival of cancer stem cells after therapy, the identification of resistance markers and the development of personalized cancer therapy. This approach is useful in the development of new therapies and personalized medicine, as it allows for the identification of patients who might benefit from a particular therapy and the prediction of resistance to therapy in other patients. With the help of molecular testing, it is possible to predict which patients will be more sensitive to certain cytostatics and which will not. In addition, the prediction of resistance to cytostatics can be useful when researching new drugs and the development of personalized medicine. Identifying molecular markers associated with resistance to chemotherapy can help develop new drugs and therapies for patients who do not respond well to existing treatments [20]. Examining cancer stem cells and predicting their resistance to therapy can be challenging and complex. Using mathematical biology models, interactions between cancer stem cells and therapeutic agents can be simulated in order to predict the effects of therapy and their resistance to it. The role of AI (artificial intelligence) in the prediction of resistance to chemotherapy in the context of CSCs is increasing and represents a potentially powerful weapon in the fight against cancer. CSCs are often responsible for resistance to chemotherapy and, therefore, the development of new therapies targeting this cell population is of critical importance.

One way to improve the therapeutic outcome is to use AI methods in the analysis of large data sets. AI can be used to identify markers that are specific to CSCs and to develop new therapies that target these cells. AI can also be used to predict the effectiveness of various therapeutic regimens in cancer patients [21]. This paper presents a GP algorithm used to develop a model for estimating the expression rates of CSC markers on cancer cells treated with 5-FU and without treatment in time. The model was trained to predict future values for 52 h (validation point) and 67 h (follow-up point). Each individual prediction model has achieved high accuracy with high coefficient determination $R^2$. The average $R^2$ was 0.96 (min. 0.93–max. 0.98) for 52 h prediction. Based on these results, we can conclude that GA can be used as a very precise auxiliary tool for in silico testing, analysis, and monitoring of the expression rate of CSC markers in cancer cells in time. The benefit of such in silico models is that, unlike experimental observations, which are discretized into discrete time intervals, they enable us to precisely monitor the state of CSC marker expression at any one time [22]. GA has a significant role in predicting resistance to chemotherapy in the context of CSCs. By using machine learning methods in the analysis of large data sets, new therapeutic targets can be identified, and personalized therapies tailored to the specific characteristics of tumors in individual patients can be developed. This opens the possibility to improve the therapeutic outcomes in patients suffering from cancer.

However, the study has several limitations that must be acknowledged. The use of a single chemotherapy agent and a limited set of CSC markers provides a focused yet narrow view of the complex interactions within the tumor microenvironment. Additionally, our study was conducted exclusively in vitro, which may not fully replicate the in vivo conditions and the systemic effects of chemotherapy on CSC populations. The predictive model developed in this study, while robust, is derived from a constrained dataset and thus requires external validation to generalize the findings. Future research should expand the spectrum of CSC markers analyzed and consider the inclusion of primary tumor samples to better account for the heterogeneity of the tumor microenvironment. Longitudinal studies involving a broader range of chemotherapeutic agents and combinations thereof could provide deeper insights into the dynamic nature of CSCs and their role in drug resistance. The clinical implications of our findings are significant. Understanding the mechanisms of CSC-related resistance to chemotherapy can guide the development of more effective therapeutic strategies. Targeting CSCs directly, possibly in combination with traditional chemotherapy, may improve treatment outcomes and prevent relapse. Moreover, the application of machine learning models, such as the genetic algorithm used in our study, demonstrates the potential of computational methods in predicting treatment responses, paving the way for personalized medicine approaches in oncology. In conclusion, while our study offers valuable insights into the CSC phenotype and its implications for therapy resistance, it also highlights the necessity for further research. By employing a multidisciplinary approach that integrates advanced technologies and personalized treatment strategies, we can aim to enhance therapeutic efficacy and patient care in the fight against cancer.

## 5. Conclusions

In conclusion, the study investigates the role and characteristics of cancer stem cells (CSCs) in chemotherapy resistance. Identified marker profiles, including CD24, CD44, ALDH1, and ABCG2, are linked to increased CSC expression post 5-FU treatment. The study underscores the importance of developing targeted CSC therapies to enhance chemotherapy efficacy and prevent tumor recurrence. Furthermore, the research emphasizes the potential of artificial intelligence, specifically genetic algorithms, in predicting CSC marker expression over time. Overall, it underscores the necessity for further research to enhance therapeutic effectiveness and patient care in the battle against cancer.

**Supplementary Materials:** The following supporting information can be downloaded at: https://www.mdpi.com/article/10.3390/curroncol31030091/s1, Figure S1: Representative dot plots of expression rate (CD44, ALDH1 and ABCG2) markers on MDA-MB-231 cell line (control group and group treated with 5-FU). Histograms detail specific marker expression up to 43 h due to observed saturation of marker expression beyond this point, which did not contribute additional insights into the treatment effects; Figure S2: Representative dot plots of expression rate (CD44, ALDH1 and ABCG2) markers on HCT-116 cell line (control group and group treated with 5-FU). Histograms detail specific marker expression up to 43 h due to observed saturation of marker expression beyond this point, which did not con-tribute additional insights into the treatment effects; Figure S3: GA decision tree for MDA-MB-231 cells—CD24- CD44+; Figure S4: GA decision tree for MDA-MB-231 cells—CD24- CD44+ + 5-FU; Figure S5: GA decision tree for MDA-MB-231 cells—ALDH1+; Figure S6: GA decision tree for MDA-MB-231 cells—ALDH1+ + 5-FU; Figure S7: GA decision tree for MDA-MB-231 cells—CD24- ABCG2+ ALDH1+; Figure S8: GA decision tree for MDA-MB-231 cells—CD24- ABCG2+ ALDH1+ + 5-FU; Figure S9: GA decision tree for HCT-116 cells—CD44+; Figure S10: GA decision tree for HCT-116 cells—CD44+ + 5-FU; Figure S11: GA decision tree for HCT-116 cells—ALDH1+; Figure S12: GA decision tree for HCT-116 cells—ALDH1+ + 5-FU; Figure S13: GA decision tree for HCT-116 cells—CD44+ ABCG2+ ALDH1+; Figure S14: GA decision tree for HCT-116 cells—CD44+ ABCG2+ ALDH1+ + 5-FU; Figure S15: MDA Control + 5-FU CD24- Abcg2+ ALDH+ i CD24- CD44+ gating strategy; Figure S16: MDA Control + 5-FU ALDH+ gating strategy; Figure S17: HCT control + 5-FU CD44+ and ALDH1+ gating strategy; Figure S18: HCT control + 5-FU CD44+ ABCG2+ ALDH1+ gating strategy.

**Author Contributions:** Conceptualization, B.L., D.C. and M.G.J.; methodology, M.Ž., D.Š., N.M.D., N.K., M.J. and I.P.; software, D.N. and N.F.; validation, A.R.H., S.N. and B.L.; formal analysis, N.F.; investigation, M.Ž.; resources, B.L.; data curation, D.N.; writing—original draft preparation, N.K.; writing—review and editing, D.C.; visualization, D.N.; supervision, M.G.J.; project administration, N.F.; funding acquisition, B.L. All authors have read and agreed to the published version of the manuscript.

**Funding:** This research received external funding by the Ministry of Science, Technological Development and Innovation of the Republic of Serbia, contract number [451-03-47/2023-01/200107 (Faculty of Engineering, University of Kragujevac), 451-03-47/2023-01/200378 (Institute for Information Technologies Kragujevac, University of Kragujevac), 451-03-9/2021-14/200378 (Faculty of Medical Sciences, University of Kragujevac)]. Junior projects of Faculty of Medical Sciences, University of Kragujevac JP 25/19, JP 05/20, JP 06/20, and JP 24/20. This work is supported by the European Union's Horizon 2020 research and innovation programme under grant agreement No. 952603 (SGABU). This article reflects only the author's view. The Commission is not responsible for any use that may be made of the information it contains.

**Institutional Review Board Statement:** Not applicable.

**Informed Consent Statement:** Not applicable.

**Data Availability Statement:** The data presented in this study are available in this article and supplementary material.

**Acknowledgments:** In this manuscript, the language model ChatGPT, (free version GPT-3.5) developed by OpenAI, was used only and exclusively for English language improvement and language support more precisely only for translation. Authors are fully responsible for the originality, validity, and integrity of the content of their manuscript.

**Conflicts of Interest:** The authors declare no conflicts of interest.

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
