# Peer review of "Modeling 5-FU-Induced Chemotherapy Selection of a Drug-Resistant Cancer Stem Cell Subpopulation"

_curroncol, doi:10.3390/curroncol31030091_

Round 1
Reviewer 1 Report
Comments and Suggestions for Authors
Hamzagic et al. describe in their manuscript about the investigation of the level of different CSC markers in various tumor cells after treatment with the well-known chemotherapeutic drug 5-FU. They compared these findings to the prediction of a genetic algorithm model, which found very similar predictions to their results. The manuscript is well-written, timely and highlights very important aspects, namely the possible resistance mechanisms of current chemotherapeutic treatments.
The title is too general and should be improved to specify the findings of the study. I suggest including the term 5-FU in the new title.
Figure 3. control bars have no SD.
Please specify in the methods that the cell lines are of human origin.
In figure 3. The control values go up from around 20 to 50 percent, it shows that even control cells started cell death for some reason. Can you explain the reason to it?
Values of control data increased over incubation time in other experiments as well (Fig 2., 7.), Percentage of (A) CD24-CD44+, 204 (B) ALDH1+ and (C) CD24-ABCG2+ALDH1+ tumor cells untreated. Can you explain the reason to it?
CD133 was not tested on HCT-166 cells, can you explain why.
Comments on the Quality of English LanguageQuality of English language of the manuscript is fine.
Reviewer 2 Report
Comments and Suggestions for Authors
Please find the attached file

Round 2
Reviewer 2 Report
Comments and Suggestions for Authors
The authors have addressed all my concerns.